# BACH2: The Future of Induced T-Regulatory Cell Therapies

**DOI:** 10.3390/cells13110891

**Published:** 2024-05-22

**Authors:** Daniel Zwick, Mai Tram Vo, Young Jun Shim, Helena Reijonen, Jeong-su Do

**Affiliations:** 1Frederick National Laboratory, Frederick, MD 21701, USA; 2School of Medicine, Johns Hopkins University, Baltimore, MD 21287, USA; 3Cardiovascular and Metabolic Sciences, Lerner Research Institute, Cleveland Clinic, Cleveland, OH 44195, USA; shimy@ccf.org; 4Department of Immunology and Theranostics, City of Hope, Duarte, CA 91010, USA; hreijonen@coh.org

**Keywords:** BACH2, T lymphocytes, inducible regulatory T cells, FOXP3 stability

## Abstract

BACH2 (BTB Domain and CNC Homolog 2) is a transcription factor that serves as a central regulator of immune cell differentiation and function, particularly in T and B lymphocytes. A picture is emerging that BACH2 may function as a master regulator of cell fate that is exquisitely sensitive to cell activation status. In particular, BACH2 plays a key role in stabilizing the phenotype and suppressive function of transforming growth factor-beta (TGF-β)-derived human forkhead box protein P3 (FOXP3)^+^ inducible regulatory T cells (iTregs), a cell type that holds great clinical potential as a cell therapeutic for diverse inflammatory conditions. As such, BACH2 potentially could be targeted to overcome the instability of the iTreg phenotype and suppressive function that has hampered their clinical application. In this review, we focus on the role of BACH2 in T cell fate and iTreg function and stability. We suggest approaches to modulate BACH2 function that may lead to more stable and efficacious Treg cell therapies.

## 1. Introduction

BACH2 is a highly conserved transcriptional repressor that controls the maturation and differentiation of T and B lymphocytes [1,2,3]. Originally, BACH2 was identified as an important molecule in B cell development. However, it was subsequently discovered that BACH2 is highly expressed in CD4 and CD8 T cells and plays a significant role in T cell differentiation to Th1 [4,5] and Th2 cells [6] and in stabilizing regulatory T cell (Treg)-mediated homeostasis.

The identification of a role for BACH2 in Treg cells may have important implications for therapeutic applications of Tregs. Tregs can be classified as thymus-derived natural FOXP3+ CD4 Treg cells (nTreg) and FOXP3+ inducible Treg cells (iTregs). nTregs are critical for the maintenance of immune homeostasis [7,8]. FOXP3+ iTregs can be generated ex vivo in the presence of IL-2 and TGF-β, exhibiting similar suppressive activity to that of nTreg [9,10]. As FOXP3+ iTregs can be readily expanded in large quantities from naïve T cells ex vivo, they have potentially enormous clinical application in autoimmune disorders and graft-versus-host disease [11]. However, the unstable phenotype and function of iTregs and their conversion to pathogenic T cells in an inflammatory environment remain major obstacles to their clinical application [11,12].

Targeting BACH2 function is a potential approach to stabilize the iTreg phenotype. Several studies have elucidated a key role for BACH2 in iTreg differentiation and function. Although the role of BACH2 in T cells has received much attention [4,13,14], the factors that regulate BACH2 expression in the development of mature CD4^+^ T-cells, as well as the mechanisms that establish a Treg-specific transcriptional program, have only recently been elucidated. This review provides an overview of the molecular mechanisms of BACH2 function and their relevance to autoimmune and inflammatory conditions. Furthermore, we highlight recent findings regarding the role of BACH2 in iTregs, with special attention to the clinical significance of BACH2 in generating stable iTregs.

## 2. BACH2 Structure

The structure of the BACH family of transcription factors has been reviewed in detail previously [15]. The BACH family consists of a basic region leucine zipper (bZip) containing transcription factors that form heterodimeric complexes with Maf proteins to repress gene expression (Figure 1). Maf proteins bind to palindromic DNA sequences termed Maf recognition elements (MAREs). BACH2 also contains an N-terminal BTB domain that mediates protein–protein interactions and a C-terminal nuclear export sequence (NES). Through experimental mutagenesis, the sites of post-translational modification have been identified, including serine phosphosites and lysine residues that undergo SUMO2 conjugation. Disease-associated mutations have been identified and linked to specific defects in BACH2 protein stability and function (Figure 1). In addition, single nucleotide polymorphisms (SNPs) that have been noted to affect protein stability have been identified. Associations with autoimmune diseases have been reported with BACH2 (See Table 1).

Much of our knowledge of the molecular mechanisms of BACH2 regulation comes from studies in B cells. BACH2 localization is regulated by a nuclear export sequence at the C terminus and a nuclear localization sequence in the bZip domain. PI3K signaling leads to BACH2 phosphorylation and promotes its cytoplasmic localization (see Figure 2). Oxidative stress inhibits the function of the cytoplasmic localization sequence and induces nuclear accumulation of BACH2 [36]. BACH2 is phosphorylated at several residues, yet only a single site appears to play a key role in controlling its nuclear localization [37].

## 3. BACH2 Cell Type-Specific Roles

### 3.1. BACH2 in T-Cell Differentiation

BACH2 plays a critical role in controlling T-cell differentiation pathways and can serve as a potent repressor of T cell polarization and functional responses (reviewed in [4]). Numerous studies have demonstrated that BACH2 can act at the transcriptional level to inhibit T-cell differentiation programs. BACH2 can integrate cytokine signals to inhibit STAT3-mediated CXCR5 expression and modulate the balance between Th17 and Tfh cell differentiation [38]. BACH2 regulates CD8 T-cell differentiation, and BACH2 deletion in naïve CD8 T cells results in changes in chromatin architecture that resemble effector and memory T cells [39,40].

### 3.2. BACH2 in iTregs: Regulation of FOXP3 Expression and iTreg Stability in Humans

BACH2 has been implicated in the stabilization of FOXP3 iTregs. The role of BACH2 in iTregs has important implications for iTreg-based cell therapeutics. Although iTregs can be readily induced in large quantities, the instability of FOXP3 expression in iTregs, both during ex vivo expansion and upon transfer to a pro-inflammatory environment, remains a critical obstacle to their clinical implementation. Thus, BACH2 could potentially be targeted for the generation of more stable iTreg-cell-based therapeutics.

Prior work has shown BACH2 regulates transcription downstream of the TCR to control Treg cell differentiation and homeostasis [41]. Murine gene deletion studies have revealed that BACH2 is a critical transcription factor for iTreg function [5,42] and represses genes associated with effector T-cell differentiation [5]. In addition, Tregs derived from mice lacking BACH2 exhibit a reduced number and diminished capacity to control T-cell-mediated colitis [5,42]. BACH2 plays a critical role in stabilizing Treg suppressor function by inhibiting T effector differentiation programs [5,6].

BACH2 is implicated in the unique gene expression profile of umbilical cord blood (UCB)-versus adult peripheral blood (AB)-derived naive CD4+ T cells [43]. The inhibition of BACH2 suppresses interleukin-2 (IL-2) expression in UCB CD4^+^CD45RA^+^ T cells [44]. However, BACH2 has been identified as an important regulator in the formation and function of CD4+ T cell lineages across various diseases [13]. It seems its role is dynamic in the status of CD4 T cells. The differential expression of BACH2 in CD4 T cells from UCB versus AB controls IL-2 production in CD4 T cells [44], regulates IL-2 receptor signaling in Tregs [45], and plays a key role in regulating Treg functions [46]. Intriguingly, UCB iTregs exhibit more robust, sustained FOXP3 expression and enhanced stability compared with AB iTregs. Mechanistically, BACH2 regulates human UCB iTreg development via direct transcriptional activity at the FOXP3 promoter [47], and BACH2 binds the FOXP3 promoter region in differentiated UCB but not AB iTregs. These studies revealed that BACH2 regulates the transcriptional mechanisms controlling FOXP3 expression in UCB iTregs, and differential expression and regulation of BACH2 could account for enhanced FOXP3 stability in UCB iTregs. Therefore, it is of interest to identify the mechanisms controlling BACH2 expression and the regulation of FOXP3 expression.

Recent insights into the mechanisms of BACH2-mediated transcriptional control of iTreg stability come from the analysis of transcription factor binding sites. Nuclear factor of activated T cells (NFAT) proteins are activated by calcium signaling pathways upon T-cell receptor (TCR) stimulation. Once activated, NFAT translocates into the nucleus, where it binds to specific DNA sequences, known as NFAT response elements, in the promoter regions of target genes [48]. FOXP3 expression and the iTreg suppressor phenotype depend on interactions with the NFAT transcription factor [49]. In FOXP3+ Treg cells, NFAT contributes to the expression of genes involved in Treg cell development and function, including FOXP3 itself. The BACH2 consensus binding sequence overlaps with that of NFAT (Figure 3). Thus, one intriguing possibility is that BACH2 competes with NFAT for the same binding sites in iTregs to regulate iTreg transcriptional stabilization. Also, FOXP3 interacts with BACH2 in a co-immunoprecipitation binding assay [47]. Because BACH2 expression is higher in UCB than AB iTregs, BACH2 regulation of FOXP3 and competition with NFAT could contribute to the differential stability of UCB and AB iTregs. In analogy to this proposed mechanism, it has been found that BACH2 modulates CD8 T-cell differentiation by regulating AP-1 binding to enhancer regions of TCR-regulated genes [3]. AP-1 is a dimeric transcription factor composed of proteins from the Jun (c-Jun, JunB, JunD) and Fos (c-Fos, FosB, Fra-1, Fra-2) families [50]. AP-1 activity is regulated by various signaling pathways, including those activated downstream of TCR engagement [51]. In addition, BACH2 represses the AP-1-driven expression of IL-2 in CD4 T cells [52]. Thus, it is of interest to determine whether an analogous mechanism of BACH2/NFAT competition is involved in the regulation of iTreg stability. In addition to competition for transcription factor binding sites, it has been identified that BACH2 can regulate NFAT expression. In particular, reduced BACH2 expression was correlated with aberrant NFAT expression in T cells derived from chronic myelogenous leukemia patients [53].

### 3.3. BACH2 in T-Cell Exhaustion

A prior comprehensive review suggested that BACH2 may be a key regulator of exhaustion [4]. Additional reports support this notion. For instance, recently, BACH2 was identified as a key regulator of exhaustion in precursor T cells in chronic viral infection [41,54] and has also been implicated in Treg quiescence [55]. In addition, BACH2 plays a critical role in stabilizing Treg naive phenotypes [5,6] and suppressing the expression of genes associated with T-cell exhaustion [39].

## 4. Molecular Mechanisms That Regulate BACH2

### 4.1. BACH2 Regulation by MicroRNAs (miRs)

Accumulating studies have highlighted the diverse mechanisms regulating BACH2 function (Figure 4). miRs are small, non-coding RNAs that bind to specific sequences in the mRNA molecules of target genes, leading to mRNA degradation or translational repression [56]. miRs play a crucial role in the regulation of BACH2 expression. miRs have been implicated in the unique transcriptional regulatory mechanisms in UCB CD4 T cells [57,58], including the regulation of BACH2 expression specifically. Interestingly, in each case, miR-mediated downregulation of BACH2 was associated with enhanced susceptibility of different cell types to apoptosis [59]. One study identified that miR 16-5p and miR 145-p promote apoptosis of human gingival epithelial cells via the downregulation of BACH2 [60]. Other miR studies have supported the critical role of BACH2 in immune cell differentiation and function. For instance, miR 148a targeted BACH2 expression to promote plasma-cell differentiation [61]. Another study identified that miR 150-5p regulates BACH2 to modulate T-cell activation in a mouse model of severe aplastic anemia [62].

miR regulation of BACH2 has been implicated in regulating the proportion of Tregs in diffuse large B-cell lymphoma [63].

### 4.2. Metabolic Regulation of BACH2

An emerging research area is the regulation of BACH2 by metabolic status. BACH2 is regulated by oxidative stress [64,65]. Several studies indicate the mammalian target of rapamycin (mTOR) signaling regulates BACH2 function [37,66] (see Figure 2 and Figure 4). mTOR signaling plays a key role in T-cell development, CD4 T-cell differentiation, and regulatory T-cell generation in vivo [67,68,69]. Among its effects on cellular functions, mTOR signaling plays a critical role in modulating mitochondrial function [70]. Interestingly, one study showed that mitochondrial function regulates BACH2, and in turn controls B cell fate outcomes [71]. Subsequent work revealed that protein kinase C (PKC) beta regulates B cell fate through mTOR signaling, metabolic programming involving mitochondrial remodeling, heme metabolism, and BACH2 inhibition [72].

In support of the importance of metabolic regulation of BACH2 in stabilizing iTreg phenotypes, mesenchymal stromal cell (MSC) co-culture during iTreg expansion stabilizes BACH2 expression and is associated with a reduced expression of markers of exhaustion and senescence [73]. The phenotype of iTregs cultured on MSCs was consistent with the core phenotype of natural Tregs (nTregs), showing positive staining for FOXP3, CD25, and cytotoxic T lymphocyte antigen-4 (CTLA4).

More generally, accumulating studies have shown that FOXP3 expression and Treg function are regulated by metabolic inputs. Fatty acid oxidation regulates Treg differentiation and function [74,75]. Further, oleic acid treatment is observed to overcome defects in suppressive function in Tregs from multiple sclerosis (MS) patients [76].

In addition to metabolism regulating Treg stability, signaling modulates metabolism in T cells, which may have consequences for BACH2 function. PD1 inhibits T-cell activation in part via inhibiting metabolic programming through the inhibition of glycolysis and fatty acid oxidation [77]. Tan et al. recently identified that PD1 restraint of Treg activation plays a role in immune tolerance, in part via PD1 inhibition of phosphoinositide 3 kinase (PI3k)–Akt signaling [78]. Thus, it is of interest to investigate whether these and other metabolic inputs may potentially affect BACH2 expression and/or function as well.

FOXP3 can also modulate metabolism, increasing oxidative phosphorylation and nicotinamide adenine dinucleotide (NAD) oxidation, which serves to adapt Tregs to low-glucose conditions [79]. In addition, mitochondrial stress during continual T-cell stimulation under hypoxic conditions, as in the tumor microenvironment, was found to drive T-cell exhaustion, and modulating T cell reactive oxygen species (ROS) could ameliorate exhaustion. Because mitochondrial function regulates BACH2 function and control of Treg suppression, mitochondrial stress in Tregs may influence BACH2 function and thereby contribute to T-cell exhaustion and the loss of suppressive function during ex vivo expansion of induced Tregs (iTregs). 

### 4.3. Regulation of BACH2 by SUMOylation and Role in iTreg Stability

Several studies have shed light on new mechanisms of BACH2 regulation and the links between metabolism, BACH2, FOXP3, and iTreg stability [73]. Previous studies identified that the small ubiquitin modifier (SUMO) sentrin-specific protease SENP3 serves as a sensor of metabolic status, particularly intracellular ROS, which regulates the nuclear translocation of SUMOylated BACH2 [80,81], the association of SUMOylated BACH2 with promyelocytic leukemia nuclear bodies, and the repression of associated transcriptional targets [65] (Figure 2 and Figure 4). During T-cell activation, increased levels of intracellular ROS induce the accumulation of SENP3 in the nucleus. In turn, SENP3 induces the deSUMOylation of BACH2, which inhibits the nuclear export of BACH2. Accumulated nuclear BACH2 in turn modulates the expression of genes that regulate FOXP3 iTreg stability [82].

Additional studies support the importance of ubiquitylation in regulating FOXP3 and iTreg stability more generally (reviewed in detail in [83]). The ubiquitin ligase Itch and deubiquitinase ubiquitin specific protease 44 (USP44) regulate FOXP3 stability [84,85]. A recent study utilizing a CRISPR screening identified additional ubiquitin-related regulators of FOXP3 and Treg stability, including USP22 and the E3 ligase Rnf20 [86]. Further, SWI and BAF proteins were identified as regulators of the maintenance of FOXP3 expression and Treg stability in mouse nTregs by CRISPR/Cas-9 screening [87].

## 5. BACH2-Associated Diseases

### 5.1. BACH2 in Murine T Cell Inflammatory Disease Models and Autoimmunity

The variety of diseases resulting from aberrant BACH2 expression and/or function highlights its important role in immune cells (Table 1). BACH2-deficient mice exhibit defects in B cell class switch recombination and Treg differentiation. Further, BACH2 deficiency or mutation results in severe autoimmunity [5,21]. BACH2 was also recently implicated in the repression of innate lymphoid cell memory in a mouse model of asthma [88]. In addition, it has been identified that BACH2 serves to suppress T follicular helper cell expansion and aberrant germinal center formation via suppressing the transcription of CXCR5 and c-Maf genes in mice [89]. In addition to global BACH2 deficiency, the impact of Treg-specific deletion of BACH2 has been examined. Further supporting a key role for BACH2 in the restraint of autoimmunity, BACH2 deficiency in Tregs leads to dysregulated type 2 allergic inflammatory responses in mice [90].

### 5.2. BACH2 in Human Autoimmune Conditions

BACH2 has been linked to susceptibility to a variety of autoimmune conditions. A new syndrome termed ‘BACH2-related immunodeficiency and autoimmunity’ (BRIDA) appears to be a monogenic primary immunodeficiency resulting from novel deleterious heterozygous point mutations in BACH2. These mutations result in reduced BACH2 expression and transcriptional repression of BLIMP in patient lymphoblastoid cells, reduced memory B cells, and systemic lupus erythematosus (SLE) symptoms [21,22]. BACH2-promoter methylation is associated with irritable bowel syndrome (IBS) [16]. A variant of BACH2 is associated with Crohn’s disease [17]. A recent genome-wide association study (GWAS) identified BACH2 as a risk locus for Addison’s disease [34]. BACH2 is also associated with protection from chronic pancreatitis [19]. Another report identified associations between BACH2 mutations that cause BACH2 haploinsufficiency and disease conditions [21]. Additionally, a report linked a BACH2 polymorphism with polyglandular autoimmunity [91].

Type 1 diabetes mellitus (T1DM) impacts approximately 10–15% of individuals diagnosed with diabetes mellitus (DM). It arises from the autoimmune destruction of pancreatic beta cells, necessitating lifelong dependence on insulin for affected patients [92]. BACH2 has been recognized for its role in regulating the survival of pancreatic beta cells through intricate crosstalk with several proteins involved in critical cellular pathways. This regulatory function involves molecular interactions that modulate key processes essential for the viability and functionality of pancreatic beta cells in T1D [27].

Vitiligo is a persistent autoimmune skin condition marked by the depletion of melanocytes, the cells responsible for melanin production [93]. While its exact cause remains elusive, experts suspect a blend of genetic, environmental, and immunological factors. Ongoing research explores the interplay between vitiligo and BACH2, a gene identified in GWAS as potentially linked to the condition. Though no direct causal relationship has been definitively established, there is a growing curiosity surrounding BACH2’s role in vitiligo’s pathogenesis, particularly regarding immune system dysregulation and autoimmune mechanisms.

MS is a chronic autoimmune disorder characterized by inflammation and damage to the central nervous system (CNS), including the brain and spinal cord [94]. Immune-mediated mechanisms, such as the activation of autoreactive T cells and the production of pro-inflammatory cytokines, are believed to play a significant role in the development and progression of MS [95]. Studies have suggested that dysregulation of BACH2 may contribute to immune dysfunction and the development of autoimmune diseases, including MS. GWASs have identified genetic variants in the BACH2 gene that may be associated with an increased risk of developing MS [96]. While the exact role of BACH2 in MS is not fully elucidated, there is growing interest in understanding its potential involvement in the pathogenesis of the disease, particularly in the context of immune dysregulation and inflammation.

Thus, BACH2 plays a central role in restraining autoimmunity (See Table 1). Overall, mutation in the BACH2 gene can disrupt immune regulation, Treg function, B-cell differentiation, and antioxidant responses, increasing susceptibility to autoimmune diseases. Understanding the mechanisms by which BACH2 mutations contribute to autoimmunity is essential for developing targeted therapies for these conditions.

### 5.3. BACH2 in Murine Tumor Models and Human Cancers

BACH2’s significance extends beyond murine tumor models, as it also plays a pivotal role in human cancers. Studies have highlighted its crucial involvement in tumor immunosuppression, where BACH2-deficient mice implanted with tumors exhibit markedly reduced tumor growth compared with their wild-type counterparts [97]. This underscores BACH2’s impact on tumor progression and the modulation of immune responses within the tumor microenvironment. In the context of cancer, BACH2 exerts its regulatory influence on TCR-induced transcriptional changes, particularly those crucial for maintaining Treg quiescence and homeostasis [41], mechanisms involved in cancer immunosuppression [55]. BACH2 has also been associated with leukemia [23]. In addition, inhibition of the BACH2 immunosuppressive function has been implicated in KRAS-associated lung cancer resistance to chemotherapy [98]. Another study revealed that BACH2 inhibits NK maturation and cell function in mice and humans, limiting the cytotoxic response to cancer [99]. BACH2 exerts multifaceted effects in tumor immunosuppression and cancer pathogenesis, influencing diverse aspects of immune cell function and tumor biology. Its complicated regulatory functions in both innate and adaptive immune responses underscore its potential as a therapeutic target for enhancing antitumor immunity and improving treatment outcomes in various cancers.

## 6. Prospects for Stabilizing iTreg Phenotype and Function via Targeting BACH2

### 6.1. Approaches to Modulating iTreg Stability

Strategies to stabilize FOXP3 expression have shown promise in enhancing the stability of the iTreg phenotype and its function. A gene editing approach that utilized homology-directed repair for targeted insertion of an enhancer was able to overcome epigenetic silencing and enforce FOXP3 expression in primary human CD4 T cells [100]. Several approaches are being pursued to enhance and preserve Treg identity and function in chronic inflammatory conditions [101]. Strategies to enhance FOXP3 expression by modulating surface molecule function through pharmacologic means have shown promise. In a graft-versus-host disease (GVHD) model, TGF-β-induced FOXP3 expression could be stabilized, and iTreg suppressor function was enhanced by co-treatment with rapamycin and IL-2/anti-IL-2 antibody complexes [102,103]. Yet, rapamycin impairs the overall efficiency of Treg expansion [104] and can promote the formation of potentially pathogenic memory T cells [105]. Notably, numerous approaches aimed at stabilizing iTreg function target epigenetic mechanisms, including binding to the FOXP3 enhancer at conserved non-coding sequence (CNS)-1, inducing the demethylation of the CNS2 of FOXP3 and the Treg-specific demethylated region (TSDR) of CD25 and inhibiting histone deacetylase (HDAC) [106]. The broad range of targets regulated by HDAC raises concerns of off-target effects of HDAC inhibitors. In light of this concern, new approaches to stabilizing iTreg function are needed.

### 6.2. Identifying Targets for Pharmacological Modulation of BACH2 Function

The multiple inputs regulating BACH2 function offer hints at avenues for therapeutic stabilization of iTreg function (see Figure 2). Potentially, small molecules modulating BACH2 function or specific related genes could be useful in this context, although more targeted approaches are needed. While there are no known direct activators or inhibitors of BACH2, the HDAC3 inhibitor RGFP966 inhibits the repressor function of BACH2 in T cells and at the Prdm1 locus in B cells [107]. In another study, BACH2 was found to be involved in the response to cytotoxic drug-induced ROS and apoptosis of B lymphoma cells. Further, PI3K inhibition attenuated the nuclear translocation of BACH2 and increased sensitization to cytotoxic drugs [80]. BACH2 has been found to regulate cell survival in the context of germinal-center B-cell development via direct repression of the pro-apoptotic mediator Bim [108]. The notion of BACH2 involvement in cell survival and stress responses has also been reported in relation to BACH2 expression in stressed islet cells in T2D. In this context, one report identified that islet-specific BACH2 knockout restores nondiabetic phenotype in human T2D islets, and BACH2 inhibition can reverse T2D in mice [109].

The identification of downstream signaling molecules involved in pathways of metabolic regulation of BACH2 function in cell stress and survival may lead to more directed therapeutics. CRISPR technology has revolutionized the discovery of novel target genes in a variety of disease contexts. A recent report confirmed HDAC1 as a key regulator of BACH2 function in plasma cell differentiation via CRISPR screening [110]. Transcriptional changes mediated by BACH2 in conventional T cells (Tconv) were examined using a luciferase reporter assay under a tetracycline-inducible BACH2 expression system [111]. While this study utilized Tconvs, it suggested the general applicability of this approach in studies of BACH2 regulation of various target genes in iTreg cells. Another study utilized high-throughput allele-specific reporter assays to prioritize a BACH2 SNP with effects on gene expression and tested it in a human T cell line while deleting the orthologous sequence in mice. This approach revealed a BACH2 SNP rs72928038 with effects on BACH2 expression and CD8 T-cell function [112]. These studies may contribute to defining the underlying mechanisms of BACH2 regulation and function, enabling the screening of compounds that may regulate BACH2.

Identifying the upstream and downstream targets of BACH2 in different disease conditions may advance the development of targeted therapies. Ongoing studies also seek to further decipher the mechanisms by which BACH2 contributes to enhanced FOXP3 stability and iTreg suppressor function.

## 7. Conclusions and Future Perspectives

BACH2 stands out as a critical regulator in immune cell differentiation and function and the maintenance of immune balance. Its intricate regulation by various factors such as microRNAs, metabolic pathways, and interactions with other transcription factors underscores its significance in the cellular processes underlying autoimmune diseases and cancer immunosuppression. Indeed, several cancers and autoimmune disease conditions have been associated with altered BACH2 expression and specific BACH2 variants. Therapeutic strategies targeting BACH2, including modulation of its function and downstream signaling pathways, offer promising avenues for stabilizing regulatory T cell phenotypes in the context of autoimmunity and cancer. Future research should focus on further unraveling the mechanisms underlying BACH2’s regulatory roles across different immune cell types and disease contexts. Interdisciplinary efforts combining basic research, translational studies, and clinical trials are crucial for unlocking the therapeutic potential of targeting BACH2, potentially leading to more effective immune-modulating therapies and improved patient outcomes.

## Figures and Tables

**Figure 1 cells-13-00891-f001:**
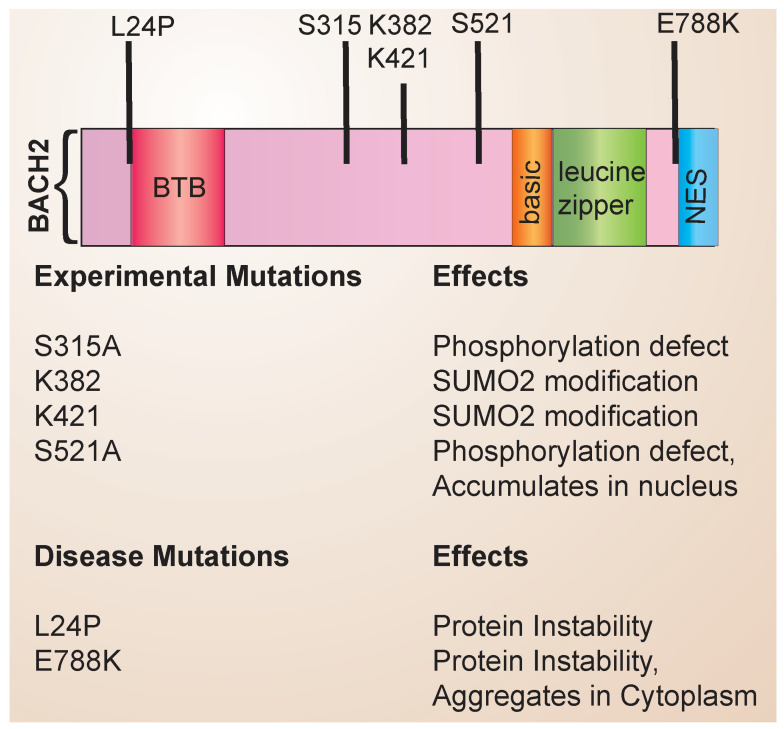
BACH2 structure and mutations. The domain structure of BACH2 is shown, along with the location and effects of known disease-associated mutations and experimental mutations.

**Figure 2 cells-13-00891-f002:**
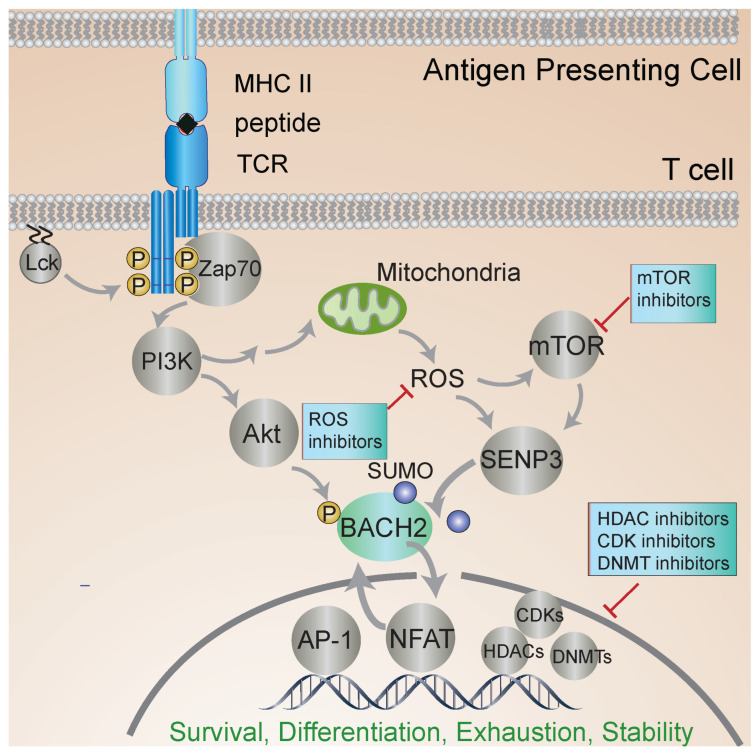
BACH2 regulates T-cell activation. T-cell receptor (TCR) engagement leads to phosphorylation of signaling molecules and the nuclear translocation of BACH2 and other transcription factors that regulate gene expression and T-cell activation. Metabolic status and other inputs regulate BACH2 localization and function. Inhibitors of the different steps of signal transduction pursued to stabilize iTreg phenotype and function are shown. The off-target effects of inhibiting HDACs and other pleiotropic factors raise the need for new, more targeted approaches. BACH2 is central to iTreg stability, regulating transcriptional responses to multiple signaling inputs. The further elucidation of the iTreg-specific roles of BACH2 should enable the identification of more targeted approaches to iTreg stabilization.

**Figure 3 cells-13-00891-f003:**
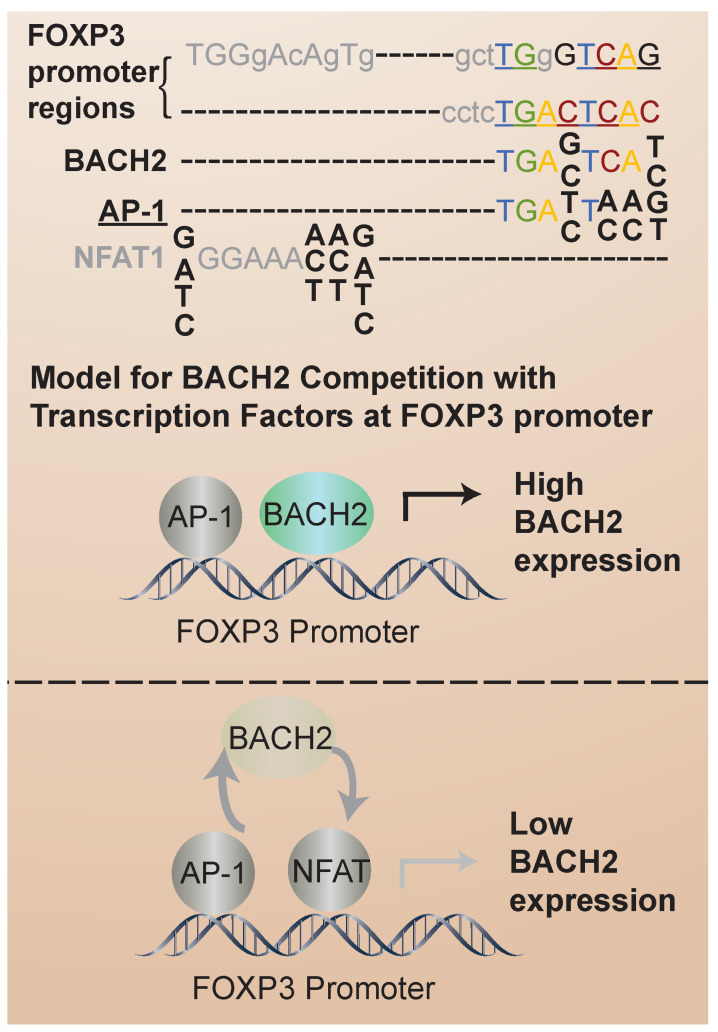
BACH2, AP-1 and NFAT consensus recognition sites in FOXP3 promoter. The overlap of BACH2 consensus binding sites with those of AP-1 (indicated by underline) and NFAT1 (indicated in gray) is shown. Competition between BACH2 and these transcription factors for binding sites in the FOXP3 promoter may regulate FOXP3 transcriptional stabilization.

**Figure 4 cells-13-00891-f004:**
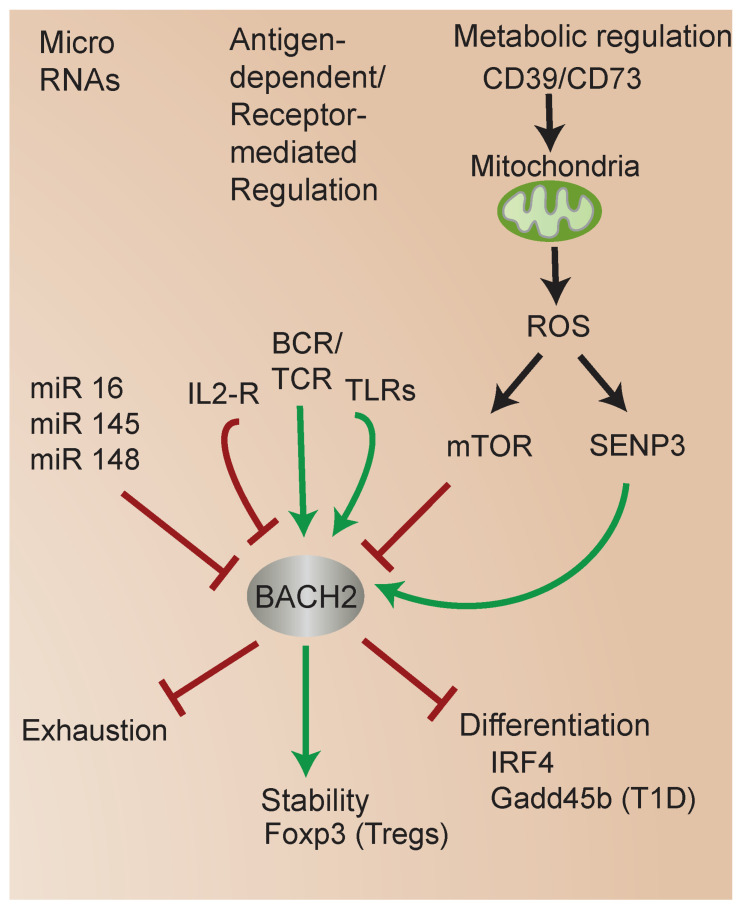
Mechanisms regulating BACH2 function. BACH2 integrates diverse metabolic and signaling inputs and mediates transcriptional responses controlling Treg stability and cell fate.

**Table 1 cells-13-00891-t001:** Conditions associated with BACH2 variants and altered functions.

Disease	BACH2 Mutation	Role of BACH2 in Disease	Reference
Inflammatory Bowel Syndrome	BACH2 promoter methylation	Affects IgG glycosylation associated with IBS	[16]
Crohn’s Disease, IBS and Non-melanoma skin cancer	BACH2 variant, SNP intron variant and genic upstream transcript variant	Associated with post-operative recurrence of Crohn’s Disease	[17,18]
Chronic pancreatitis	BACH2 associated with protection	BACH2 repression associated with features of advanced disease	[19]
Systemic lupus erythematosus	SNP rs597325	Associated with SLE	[20]
BRIDA	T71C, G2362A G1727T	BACH2 related immunodeficiency	[21,22]
Leukemia	BACH2 associated with incidence, genome wide association studies, and SNPs	Low expression associated with poor overall survival	[23,24]
Asthma		SNP associated with disease risk	[25]
Type 1 Diabetes (T1D)			[26,27]
Crohn’s and Celiac’s disease			[28,29]
Vitiligo			[30]
MS			[31,32]
Rhumatoid Arthritis			[33]
Addison’s Disease			[34,35]

## Data Availability

No data were used to support the findings of this study.

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
