# Peer review of "BACH2: The Future of Induced T-Regulatory Cell Therapies"

_cells, 2024, doi:10.3390/cells13110891_

Round 1
Reviewer 1 Report
Comments and Suggestions for Authors
Zwick D and co-workers reviewed the roles of BACH2 in T cell fate and iTreg function and stability, and proposed approaches to modulate BACH2 function that may lead to more stable and efficacious Treg cell therapies, which is clinically significant.
Major comments
1. Figure legends.
a. Need figure legends for Figures 1, 3, and 4. Figure legend should provide information that readers can understand without referring to the text.
b. Some information in the figure should also be presented in the text. For example, figure 1 shows disease mutations without mentioning them in the text.
2. Figure 2. There is nowhere in the text that Figure 2 was quoted.
3. Figure 4.
a. Figure 4 should be quoted properly in sections 4.2. and 4.3.
b. CRTC2 in Figure 4 should be mentioned in the text. If not mentioned in the text, it should be removed from Figure 4.
4. Lines 118-135: Authors proposed possible mechanisms of BACH2 of mediating transcriptional control of iTreg stability by interacting with NFAT and AP-1. Thus, the authors need to provide information about what NFAT and AP-1 do.
5. It will be helpful if the authors provide a model figure for section 6.
Minor comments
1. Line 3: For the senior author and correspondence, please use “4, *”, not “4 and *”
2. Line 105-106:
a. Change “Bach2” to “BACH2” to be consistent.
b. The sentence does not read as a correct English.
3. Line 137: change "AP1” to “AP-1” to be consistent.
4. Table 1: row 4, remove “.” In front of G1727T.
Comments on the Quality of English Language
Little edit needed.
Reviewer 2 Report
Comments and Suggestions for Authors
The study is well designed and performed, but the following points need to be addressed:
1 I think that Figure 4 is not clear. It should be revised specifying what are the inhibition mechanisms that result in BACH2 downregulation.
2 Systemic lupus erythematosus and rheumatoid arthritis should be included in Table 1.
3 The role of BACH2 mutations in the development of autoimmunity diseases and cancer should be better described.
4 Table 1 should be better organized. For instance, the table could include a column reporting the immune diseases, a column reporting BACH2 mutation, a column reporting the role of BACH2 in disease development and a column reporting the references.
Comments on the Quality of English Language
Minor editing of English language required.
Reviewer 3 Report
Comments and Suggestions for Authors
This paper provides an objective review of BACH2 immunoregulatory functions focusing its role in Treg functions and induced Treg (iTreg) stability. The authors address the perspectives for modulating BACH2 activity to stabilize iTreg cells, what is crucial for their potential use in Treg cell therapies. The paper is well written, concisely covers the main issues related to BACH2 molecular structure and biological functions, its association with autoimmune diseases and the emerging strategies for modulating BACH2 activity to stabilize iTreg cells aiming its therapeutical use. The figures are very helpful for a better understanding of the text. The authors provided an extensive and up-to-date bibliographical revision. The usefulness of this paper for a general audience is to call attention on the potential use of iTreg based therapies.
Minor comment: in line 307, substitute Cleland for Cleveland.
Round 2
Reviewer 2 Report
Comments and Suggestions for Authors
The authors answered all the raised queries and improved the manuscript.